# Statistical Analysis of Absenteeism in a University Hospital Center between 2007 and 2019

**DOI:** 10.3390/ijerph191912966

**Published:** 2022-10-10

**Authors:** Charlène Millot, Bruno Pereira, Sophie Miallaret, Maëlys Clinchamps, Luc Vialatte, Arnaud Guillin, Yan Bailly, Ukadike Chris Ugbolue, Valentin Navel, Julien Steven Baker, Jean-Baptiste Bouillon-Minois, Frédéric Dutheil

**Affiliations:** 1Occupational and Environmental Medicine, CHU Clermont-Ferrand, F-63000 Clermont-Ferrand, France; 2Biostatistics Unit, CHU Clermont-Ferrand, F-63000 Clermont-Ferrand, France; 3Blaise Pascal Mathematics Laboratory, Mathematics Department, Université Clermont Auvergne, F-63000 Clermont-Ferrand, France; 4Cikaba—Quality, Health, Safety and Environment Software for Prevention, F-63000 Clermont-Ferrand, France; 5CNRS, LaPSCo, Physiological and Psychosocial Stress, Occupational and Environmental Medicine, CHU Clermont-Ferrand, Université Clermont Auvergne, WittyFit, F-63000 Clermont-Ferrand, France; 6School of Health and Life Sciences, Institute for Clinical Exercise & Health Science, University of the West of Scotland, South Lanarkshire, Scotland G72 0LH, UK; 7CNRS, INSERM, GReD, Ophthalmology, CHU Clermont-Ferrand, University Hospital of Clermont-Ferrand, F-63000 Clermont-Ferrand, France; 8Department of Sport, Physical Education and Health, Centre for Health and Exercise Science Research, Hong Kong Baptist University, Hong Kong, China

**Keywords:** hospital, absenteeism, compressible absences, occupational factors, sociodemographic factors

## Abstract

Objectives: To estimate the evolution of compressible absenteeism in a hospital center and identify the professional and sociodemographic factors that influence absenteeism. Method: All hospital center employees have been included over a period of twelve consecutive years (2007 to 2019). Compressible absences and occupational and sociodemographic factors were analyzed using Occupational Health data. Since the distribution of the data did not follow a normal distribution, the number of days of absence was presented as a median (interquartile range (IQR): 1st quartile–3rd quartile), and comparisons were made using non-parametric tests followed by a negative binomial model with zero inflation (ZINB). Results: A total of 16,413 employees were included, for a total of 2,828,599 days of absence, of which 2,081,553 were compressible absences (73.6% of total absences). Overall, 42% of employees have at least one absence per year. Absent employees had a median of 15 (IQR 5–53) days of absence per year, with an increase of a factor of 1.9 (CI95 1.8–2.1) between 2007 and 2019 (*p* < 0.001). Paramedical staff were most at risk of absence (*p* < 0.001 vs. all other occupational categories). Between 2007 and 2019, the number of days of absence was multiplied by 2.4 (CI95 1.8–3.1) for administrative staff, 2.1 (CI95 1.9–2.3) for tenured, 1.7 (CI95 1.5–2.0) for those living more than 12 km from the workplace, 1.8 (CI95 1.6–2.0) among women, 2.1 (CI95 1.8–2.6) among those over 50 years of age, 2.4 (CI95 1.8–3.0) among “separated” workers, and 2.0 (CI95 1.8–2.2) among those with at least one child. Conclusions: Paramedical personnel are most at risk of absenteeism. Meanwhile, absenteeism is increasing steadily, and overall, the increase is major for administrative staff. The profile of an employee at risk of absenteeism is a titular employee, living at distance from work, probably female, over 50 years old, separated, and with children. Identifying professionals at risk of absenteeism is essential to propose adapted and personalized preventive measures.

## 1. Introduction

Absenteeism at work can be defined as “any unexpected absence of an employee from his or her workstation” [1,2,3]. It is a major economic and public health issue in the hospital sector [4]. A hindrance to productivity [5] and a threat to the balance of work teams [6], it is also likely to affect the quality of care [7,8]. Within healthcare facilities, caregivers have been identified as being particularly at risk for absenteeism [9]. However, studies on the subject are limited [9]. Most of the literature on absenteeism in the hospital sector focuses solely on caregivers [10,11,12,13]. Among them, caregivers’ aides are reported to have the highest absenteeism rate [9]. Furthermore, to our knowledge, there is no literature that examines the professional status or home-to-work distance as determinants of absenteeism. A few studies have also looked at the sociodemographic determinants of absenteeism [14,15,16,17,18]. Among the factors that have been widely studied are gender and age: absenteeism is thought to be higher among women [14,15,16] and older individuals [17]. The literature has also identified celibacy as a protective factor in absenteeism [10,18]. On the other hand, the majority of studies that have looked at absenteeism from work have focused on short-term or health-related absences only [9,17]. In addition, publications dealing specifically with absenteeism in healthcare institutions are generally limited to a few years of study only [7,8,9,10]. Thus, there is no literature dealing with a large volume of employees followed over the long term (several consecutive years) and incorporating a minimum of information on occupational characteristics (occupation, status, distance between home and work) and sociodemographic characteristics (age, sex, marital status, children).

The main objective of this study was to study the evolution of absenteeism among staff at a University Hospital Center over a 12-year period. Our secondary objectives were to identify the professional and sociodemographic factors influencing this absenteeism.

## 2. Method

### 2.1. Study Design

Using occupational health data in relation with Human Resources, we conducted a longitudinal study on absenteeism among all staff at the Clermont-Ferrand University Hospital. The data analyzed covered a period of twelve consecutive years, from 2007 to 2019. This study was approved by the South-East VI Personal Protection Committee and the French National Commission for Information Technology and Civil Liberties (CNIL). In agreement with the CNIL, a unique 6-digit identifier was assigned to each university hospital center employee to ensure the anonymity of data.

### 2.2. Study Population: Eligibility Criteria

The criteria for inclusion were to be part of the staff of the University Hospital Center of Clermont-Ferrand between 2007 and 2019, regardless of profession or establishment, and to benefit from annual leave. No exclusion criteria were applied to this study.

### 2.3. Judging Criteria

#### 2.3.1. Primary Judgement Criteria

So-called “compressible” absences were studied, i.e., those for which mitigation measures could be implemented because they were partly related to working conditions or health status: ordinary sick leave, occupational injury, occupational disease, strike, long-term leave, long-term sick leave, commuting accident and unauthorized absence [19]. Each absence detailed the start date, end date, reason, and duration. Non-compressible absences (annual leave, safety rest, training, union activities, maternity leave, family events, miscellaneous absence authorizations and unpaid absences) were not studied.

#### 2.3.2. Secondary Judgement Criteria

The occupational variables were occupational status (titular, non-titular), home-to-work distance (calculated from each employee’s postal code and place of work), and occupation. Occupations were studied independently and were grouped into caregiver/non-caregiver groups and then into categories (medical, paramedical, administrative, technical).

The sociodemographic variables available for each absence were: gender, age of subject, marital status, parental status, and number of children.

## 3. Statistics

Statistical analysis of absenteeism data was conducted using Stata software (v16, College Station, TX, USA). The proportions of absent employees were compared for each factor using a Chi-square test. A Shapiro–Wilks test verified that the number of days of absence did not follow a normal distribution (Appendix A). Among the absent employees, the number of days of absence was expressed as a median (interquartile range (IQR): 1st quartile–3rd quartile) and compared according to the different variables (professional and sociodemographic) via the non-parametric tests of Mann–Whitney (2 groups: gender, titular/non-titular, caregiver/non-caregiver, home–work distance, parentality) and Kruskal–Wallis (>2 groups: occupation, age, marital status). If there were more than two groups, “multiple pair-wise comparison tests” (paired comparison) were conducted. Effect sizes were also calculated between each group. Age classes were carried out according to quartiles: <30 years, 30–40 years, 40–50 years, and >50 years. Home–work distances were separated into two classes according to the median: less and more than 12 km. The risk of absence as a function of each variable (professional and sociodemographic) was calculated using a negative binomial model with zero inflation (ZINB), considering the high proportion of 0 (zero compressible days of absence for many employees). Vuong and likelihood ratio tests where alpha = 0 were automatically performed after each negative binomial model with zero inflation to determine whether the model was appropriate. Furthermore, a sensitivity analysis was performed excluding people with 365 or 366 days of absence per year. We can therefore rely on the results of our model. The risk calculations were carried out globally (covering the 12 years of the study as a whole) and by period (2007 to 2010, 2011 to 2014, and 2015 to 2019). The three periods were defined statistically based on observed changes in the number of days of absence on the global model. The weight of each factor was studied using multivariate analyses: logistic regressions for the proportions of absentees and the ZINB model for the number of days of absence. The normality of the residuals after the models was checked using qqplot and Shapiro–Wilk tests. Risk calculations were also carried out using classical logistic regression to check the consistency of the results (sensitivity analysis). The results were expressed as: coefficient, 95% confidence interval (95% CI). For all the tests performed, the significance threshold was set at 0.05.

## 4. Results

### 4.1. Description of the Population

Over the last 12 years, 16,413 subjects were included: 4385 (26.7%) men and 12,028 (73.3%) women. The mean age was 40.3 ± 11.6 years. Caregivers represented most of the workforce (70.4%). The professional categories were mainly paramedical (43.0%), which was followed by medical (27.4%), technical (15.3%), and administrative (15.3%). More than half of the staff (58.5%) were permanent employees.

### 4.2. Study of Absenteeism—Overall Model and by Time Period

Out of a total of 2,828,599 days of absence during the twelve years studied, 2,081,553 were compressible absences (73.6% of total absences). Regular sick leave accounted for the largest number of days of absence (52.0%), followed by occupational diseases (20.7%), accidents at work (10.0%), long-term sick leave (9.3%), long-term leave (3.6%), unauthorized absences (2.1%), commuting accidents (1.3%) and strikes (0.9%). Overall, 42% of the University Hospital Center employees had at least one compressible day of absence per year, with a median of 15 (5–53) days of absence/employee/year. The proportion of absent employees remained globally stable over time, and the number of days of absence was multiplied by 1.9 (CI95 1.8–2.1) in twelve years (*p* < 0.001) (Figure 1, Figure 2 and Figure 3).

### 4.3. Professional Characteristics

Caregivers/Non-Caregivers: Caregivers were more likely to have at least one absence (44% vs. 39% for non-caregivers, *p* < 0.001, effect size 0.03) (Table 1), with an increased risk of absence of 27% (CI95 23–30%) (*p* < 0.001). Over the whole period 2007–2019, the number of days of absence of a caregiver was 1.19 (CI95 1.14–1.23) times that of a non-caregiver (16, IQR 5–55 vs. 13, IQR 4–47 days/employee/year, *p* < 0.001) (Table 1). However, the gap between caregivers and non-caregivers was no longer significant after 2015 (Table 2, Figure 2). Between 2007 and 2019, absences were multiplied by 1.8 (CI95 1.6–2.0) among caregivers and 2.4 (2.0–2.9) among non-caregivers (Figure 2).

Professional status: Tenured had a higher proportion of absent employees (27% vs. 23% among non-tenured, *p* < 0.001, effect size 0.32) (Table 1). The risk of absence was reduced by 56% (CI95 49–62%) among non-tenured and in particular by 69% over the last period (Figure 4). The number of days of absence for a titular was 2.76 (CI95 2.65–2.87) times that of a non-titular (19, 5–66 days/employee/year vs. 8, 3–24 days/employee/year for non-titular patients, *p* < 0.001) (Table 1). Between 2007 and 2019, the number of days of absence was multiplied by 2.1 (CI95 1.9–2.3) for tenured (Figure 2). The gap between tenured and non-tenured increased over time (*p* < 0.001) (Table 2, Figure 2 and Figure 4).

Occupational category: Paramedical staff had the highest proportion of absent employees (46% vs. 35% for medical staff, 37% for administrative staff, and 42% for technical staff, *p* < 0.001) (Table 1). The risk of absence was increased by 36% (CI95 32–40%) compared to medical staff. Paramedics were absent longer than other professional categories (19, 6–64 vs. 8, 2–24 days/employee/year for medical staff, 12, 3–42 for administrative staff, and 15, 5–53 for technical staff, *p* < 0.001). The number of days of absence for paramedical staff was 2.31 (CI95 2.18–2.44) times that of medical staff (*p* < 0.001) (Table 1). Between 2007 and 2019, the number of days of absence was multiplied by 1.7 (CI95 1.6–1.9) for paramedical staff, 2.4 (1.8–3.0) for administrative staff and 2.1 (1.9–2.4) for technical staff. The number of days absent increased significantly for administrative staff over time (Table 2, Figure 2 and Figure 4).

Occupation: Hospital services officers was the occupation with the highest number of absent employees (54% vs. 50% for assistant nurses, 46% for administrative officers, or 31% for hospital practitioners, for example, *p* < 0.001) (Table 1). For example, the risk of absence was increased by 67% (CI95 62–72%) compared to a hospital practitioner. Assistant nurses and hospital services officers were the two professions with the longest absences (20, 7–64 days/employee/year and 20, 7–68 days/employee/year, respectively; *p* < 0.001 vs. all other professions). For example, the number of compressible absence days for a hospital services officer was 2.44 (CI95 2.11–2.81) times that of a hospital practitioner (*p* < 0.001) (Table 1). Between 2007 and 2019, hospital services officers increased their number of days of absence by a factor of 1.9 (CI95 1.4–2.3) and assistant nurses 1.8 (1.5–2.2) (Table 2, Figure 2). The first period stands out with higher risks of absence for all occupational categories (Figure 4).

Home–work distance: This variable was reported for 1,497,957 absences (87%). Employees living more than 12 km from work were the most absent (44% vs. 40% of employees living less than 12 km from work, *p* < 0.001, effect size 0.05) (Table 1), with an increased risk of absence of 24% (CI95 20–28%). The number of days of absence of an employee living more than 12 km away from work was 1.04 (1.00–1.09) times that of an employee living less than 12 km away (17.5–57 vs. 14.4–45 days/employee/year, *p* < 0.001) (Table 1). Between 2007 and 2019, the number of days of absence was multiplied by 1.7 (CI95 1.5–2.0) among employees living more than 12 km from their place of work (Table 2, Figure 2).

#### Sociodemographic Characteristics

Gender: Women were the highest proportion of employees absent (43% vs. 36% of men, *p* < 0.001, effect size 0.09) (Table 1). Being a woman increased the risk of absence by 34% (31–38%). The number of days of absence for women was 1.12 (1.07–1.17) times that of men (15, 5–52 d/employee/year for women vs. 11, 3–38 d/employee/year for men) (*p* < 0.001) (Table 1). In particular, over the first period, it was 1.20 times higher (Figure 4, Appendix A). Between 2007 and 2019, absences were multiplied by 1.8 (CI95 1.6–2.0) for women (*p* < 0.001) (Figure 2).

Age: Those over 50 years of age had a higher proportion of absent employees than those under 30 years of age (44% vs. 40%, *p* < 0.001) (Table 1), with an increased risk of absence of 13% (CI95 8–18%, *p* < 0.001). Those over 50 years of age were absent the longest (18, 5–66 vs. 10, 3–35 for those under 30, 15, 5–48 days/employee/year for the other two age groups, *p* < 0.001), with the number of days absent 2.60 (2.48–2.74) times that of those under 30 (*p* < 0.001) (Table 1). Over the period 2007–2019, the number of days of absence was multiplied by 2.1 (CI95 1.8–2.6) among those over 50 years of age, with a widening over time of the gap between the youngest and oldest employees (Table 2, Figure 3 and Figure 4), reaching a number of days of absence 2.89 times higher over the last period (Appendix A).

Marital status: Separated employees (separated, divorced, or widowed) had the highest proportion of absent employees (29% vs. 24% for singles and 26% for couples, *p* < 0.001) (Table 1), with an increased risk of absence of 39% (CI95 35–44%) compared to singles. The number of days of absence for a separated employee was 1.28 (1.22–1.36) times that of an employee in a couple and 2.03 (1.91–2.16) times that of a single employee (20, 7–69 vs. 17, 5–59 days/employee/year for employees in a couple and 10, 3–31 days/employee/year for singles (*p* < 0.001) (Table 1). While overall between 2007 and 2019, the absences of separated employees were multiplied by 2.4 (CI95 1.8–3.0), the gap with singles increased over time (Table 2, Figure 3 and Figure 4).

Parenting: Employees with at least one child had the highest proportion of absent employees (44% vs. 39% among employees without children, *p* < 0.001, effect size 0.16) (Table 1), with an increased risk of absence of 18% (CI95 14–22%). The study results by time period were similar to these only between 2007 and 2010. After that, there was no difference between the two groups (Figure 3). The number of days of absence of an employee with at least one child was 1.48 (CI95 1.43–1.54) times that of an employee without children (16, 5–52 vs. 12, 4–38 d/employee/year) (*p* < 0.001) (Table 1). Between 2007 and 2019, the number of days of absence for employees with children was multiplied by 2.0 (CI95 1.8–2.2), with an increase in the gap with employees without children over time (Table 2, Figure 3 and Figure 4), reaching 1.74 times higher in the latter period (Appendix A).

A sensitivity analysis conducted without those with 365 or 366 days of absence showed that the significance of the factors in the ZINB models did not change except for being a caregiver or not and for the distance variable. Indeed, being a caregiver becomes a protective factor in the second period, whereas it was a risk factor before, and living more than 12 km away remains protective but becomes very significant in the overall period as well as in the third period. These are the only variables impacted during this sensitive analysis; all others show similar results.

### 4.4. Study of Compressible Absenteeism—Multivariate Analyses

Proportion of employees with at least one absence: Risk factors for absence were being a caregiver (coefficient = 0.42, CI95 0.37–0.47), paramedical (0.35, 0.31–0.40), female (0.13, 0.11–0.16), and distance to work (0.03, 0.01–0.04) (*p* < 0.001). Protective factors were being non-titular (−0.26, −0.29–−0.23) and administrative (−0.26, −0.29 −0.22), age (−0.03, −0.05–−0.02), and being in a couple (−0.11, −0.15–−0.07) (*p* < 0.001) (Figure 5).

Number of compressible absence days: Risk factors for high absence duration were being a caregiver (coefficient = 0.33, CI95 0.22–0.45), paramedical (0.25, 0.15–0.35), female (0.24, 0.18–0.30), age (0.20, 0.20–0.30), and home-to-work distance (0.10, 0.04–0.10) (*p* < 0.001). Protective factors were non-titular (−0.54, −0.61–−0.47), administrative (−0.37, −0.46–−0.29), and being in a couple (−0.16, −0.27–−0.06) (*p* < 0.001) (Figure 5).

Study of compressible absenteeism—sensitivity analyses: The results were broadly similar using a classical logistic regression model. The risk of absence increased for caregivers by 11% (CI95 10–13%), paramedics by 10% (7–14%), Hospital Services Officers by 77% (65–90%), and tenured by 33% (31–35%), employees living more than 12 km from work by 11% (9–13%), women by 19% (17–21%), over-50s by 9% (7–11%), separated employees by 18% (13–23%), and employees with at least one child by 14% (12–16%) (Appendix A). The results of period logistic regressions were also similar to the results obtained by the ZINB model (Appendix A).

## 5. Discussion

The main results showed:❖A high prevalence of absences and an increase in absenteeism over time for most of the groups studied.❖Paramedical personnel remain particularly at risk of absences even if new absentees emerge (administrative staff).❖The involvement of sociodemographic factors in the occurrence of compressible absences.

### 5.1. Prevalence of Absences and Evolution

We found that over the last 12 years, the proportion of employees absent at least once a year for a compressible reason was 42%. The median duration of absence was 15, 5 to 53 days/employee/year. In addition, we found a 1.9-fold increase (CI95 1.8–2.1) in the number of days of compressible absence between 2007 and 2019. To our knowledge, and as mentioned in the introduction, no publication to date has focused on compressible absences from work, making it impossible to compare our results with data from the literature. In fact, most of the available studies on the subject have focused on short-term absences or absences for health reasons only and have taken place over short periods of time [10,13,18]. In recent years, hospital reforms have multiplied [4]. From activity-based pricing to internal hospital changes, these reforms have contributed to the deterioration of working conditions [4]. In addition, job dissatisfaction, low decision latitude, or lack of time and resources to perform the tasks required have been associated with a high risk of absence [20,21,22]. All these factors could partly explain the observed increase in compressible absenteeism. In addition, absenteeism is associated with a significant loss of resources within healthcare institutions [4], which is itself partly responsible for the lack of available human and material resources. It can then lead to an overload of work and subsequently to new absences [21].

### 5.2. Occupational Risk Factors

We identified paramedical staff as having the highest number of compressible days of absence for the entire study period, with a median of 19, 6 to 64 days of absence per employee per year. These results are consistent with those in the literature [9,23]. It is important to note, however, that the available studies cover all reasons for absence and not just compressible absenteeism. These results can be explained in part by the high rate of job dissatisfaction [24,25,26,27,28,29]. A 2008 survey of job satisfaction among caregivers in various countries, including France, revealed that more than 40% of those surveyed were dissatisfied with their physical working conditions and psychological support [25]. At the beginning of our study period, administrative staff were at low risk of absence. We then noted an alarming increase in compressible absence days within this occupational category. Since this appears to be a new phenomenon, we did not identify any available data on the subject. We can, however, try to explain this phenomenon by the growing job dissatisfaction of all hospital staff due to the successive reforms of recent years [4]. Caregivers, in particular, are at very high risk of job stress and burnout [30,31,32,33]. Burnout is indeed increasingly present in hospitals and can even lead to suicide among some professionals [34]. The last major occupational risk factor was living far from the workplace. Repetitive commuting can in fact contribute to increasing fatigue and stress levels. In addition, most trips are made by car, which increases sedentariness and has been identified as one of the factors reducing work capacity [35,36].

### 5.3. Sociodemographic Factors and Absenteeism

We first identified women as having a higher number of compressible days of absence per year. This finding is consistent with the data in the literature [37,38]. Some publications have attempted to find an explanation for this discrepancy and in particular have shown that women have more difficulty balancing work and private life [16]. Another explanation would lie in the health status and personality differences between men and women [38]. Then, we determined that employees in the oldest age group had the highest number of days of absence per year. These results are consistent with those in the literature [17,39]. This finding seems easily explained insofar as the state of health of the individual, identified as a determinant of absenteeism, is closely linked to age. We were able to identify that being single was a protective factor for compressible absences. On the other hand, having at least one child was identified as a risk factor for compressible absences. These results are again similar to the data in the literature [10,16,18]. All of these results can be explained in part by the difficulty of reconciling work and family life [16].

## 6. Limitations

The main limitation of our study was the use of imperfect statistical models. Indeed, we used a negative binomial model with zero inflation, which is usually used to process count data with a high proportion of zeros. However, our data did indeed present a high proportion of employees without compressible absences but also employees with many days of absence (365 days in the year). However, these individuals were too few in number for the model to give false results (Appendix A). Indeed, a sensitivity analysis was performed without those individuals with a peak of 365 or 366 days of absence. The results were not different from those found previously. The models used also did not allow the study of absence kinetics. The second limitation of our work was the missing data. However, these only concerned the variable “working distance” and represented only 10% of the data. Thus, we still had access to data on more than one million absences, which underlines the quality of the database on which our work is based. For reasons of anonymity, we were unable to access each employee’s work service and details of the reasons for absence. Access to these data as well as to activity indicators (such as patient data) could be relevant for future studies on the subject. Indeed, an increase in the volume of activity grouped with a lack of staff (linked to reforms and budget restrictions) could be one of the explanations for the increase in absenteeism in hospitals. Finally, compressible absences are by definition absences for which mitigation measures can be put in place [40,41,42,43,44,45,46]. It would be relevant to cross-reference the results obtained in our study with information on the preventive measures already adopted by the hospital center to assess their effectiveness. Finally, our study was retrospective, and it was therefore impossible to take into account certain professional subjective variables such as job satisfaction, social support or psychological load.

## 7. Conclusions

Absenteeism in healthcare institutions is a public health issue insofar as it can lead to an alteration in the quality of care. Compressible absenteeism, which accounts for more than 2/3 of absences from the university hospital center, has been on the rise for several years. While compressible absenteeism affects all the professions at the university hospital center, some remain more at risk than others, particularly hospital services officers and assistant nurses. Some professionals are more exposed to the risk of absenteeism (paramedical and technical fields), while others, who seemed protected ten years ago, are experiencing a worrying increase in the prevalence of absenteeism (administrative staff). Conversely, non-tenured staff are experiencing a decrease in risk compared to tenured staff. While professional characteristics seem to play a major role in the occurrence of absences, particularly in relation to working conditions, it is important to highlight the importance of sociodemographic factors. Indeed, we found that being a woman, being a parent, being elderly or being separated significantly increases the risk of absenteeism. Reducing compressible absences is now a major challenge for human resources and actions exist to achieve this. Among the means explored are health promotion actions (physical activity and nutrition) or actions to improve the work environment (communication, conflict management) [42,43,44,45,46]. These measures seem to show promising results and could make it possible to limit absences and the associated costs.

## Figures and Tables

**Figure 1 ijerph-19-12966-f001:**
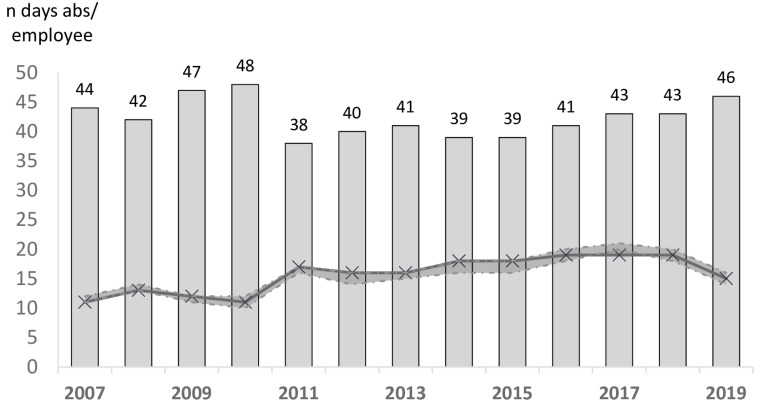
Evolution in the proportion of absentees (the crosses) and the number of days of absence (the bars) per absent employee (expressed in median ± one centile).

**Figure 2 ijerph-19-12966-f002:**
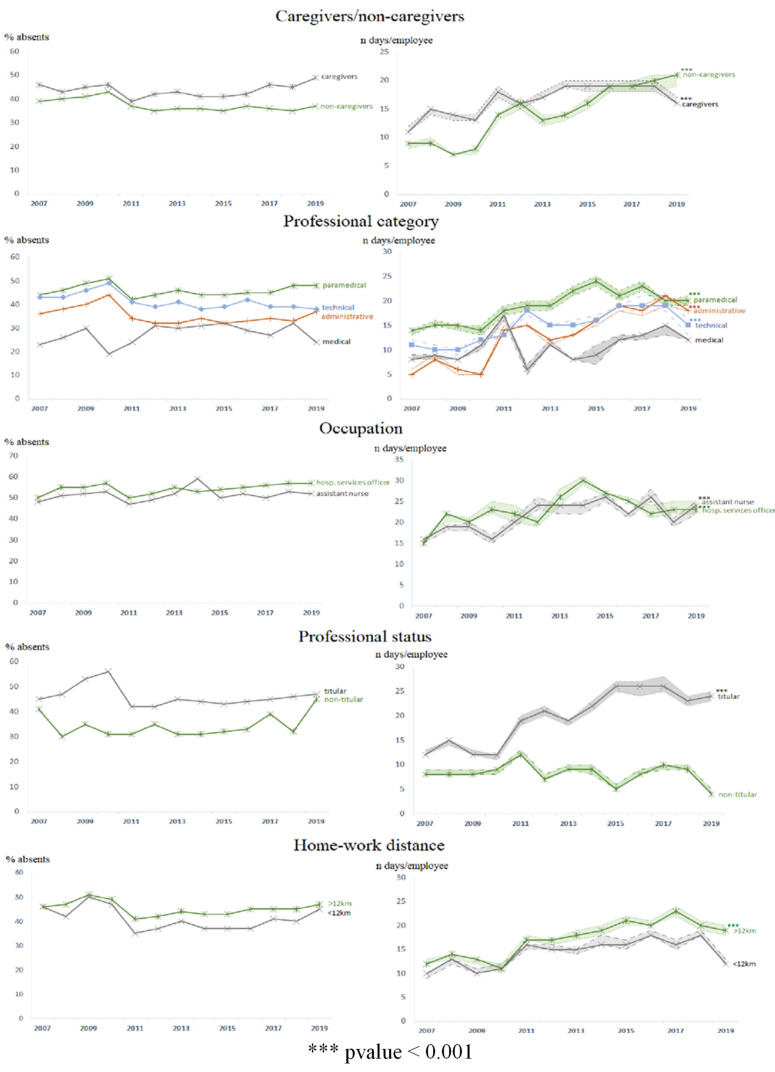
Evolution in the proportion of absentees and the number of days of absence per absent agent by occupational factor (expressed in median (C49–C51)).

**Figure 3 ijerph-19-12966-f003:**
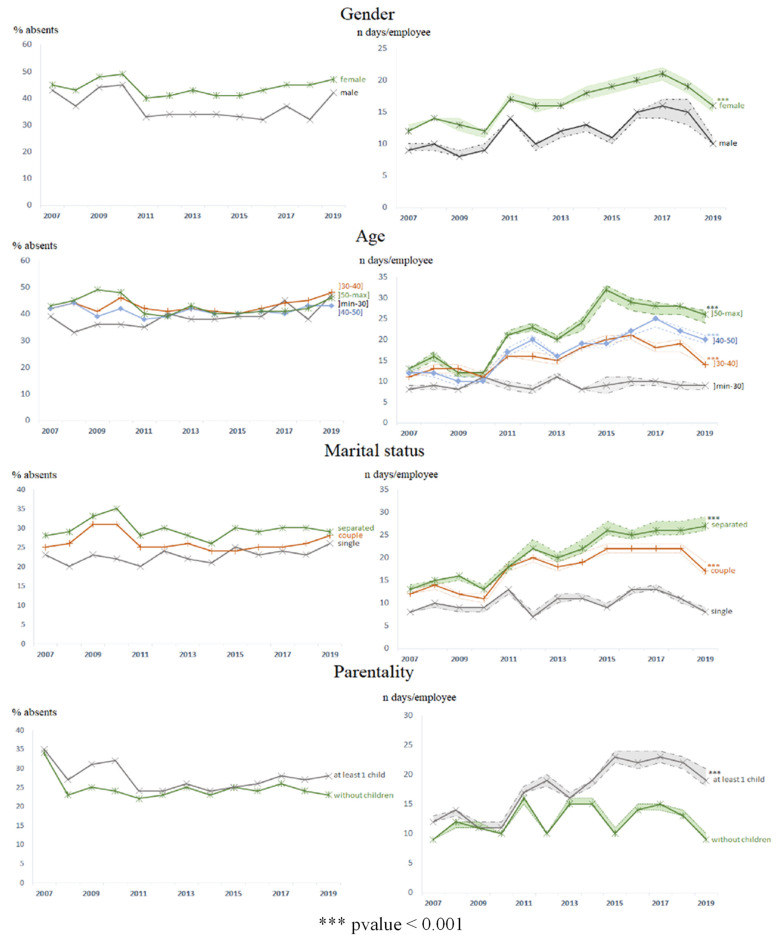
Evolution of the proportion of absentees and the number of days of absence per absent agent by sociodemographic factor (expressed in median (C49–C51)).

**Figure 4 ijerph-19-12966-f004:**
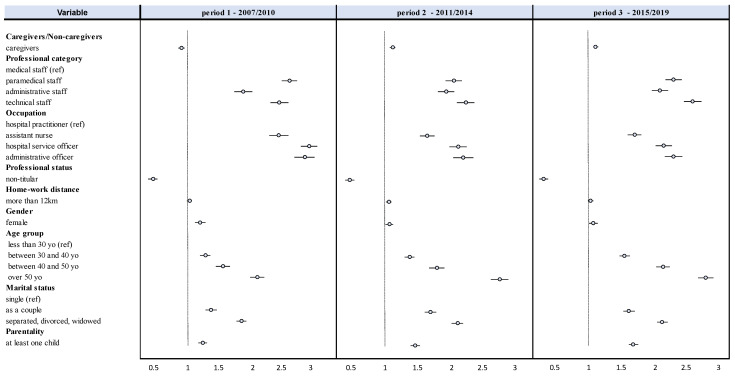
Forest plot of the incidence rate ratio of compressible absence risk as a function of occupational and sociodemographic factors by period (zero inflated negative binomial model (ZINB)).

**Figure 5 ijerph-19-12966-f005:**
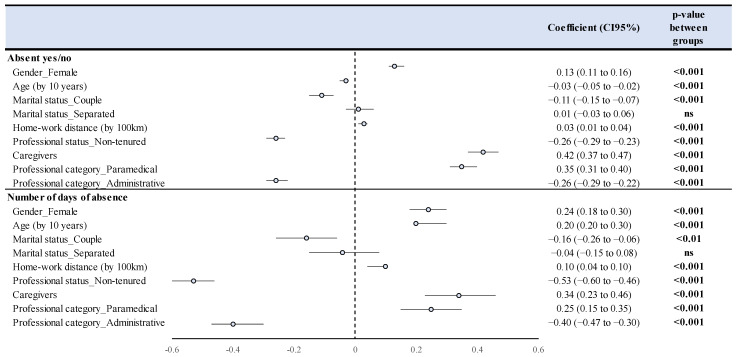
Multivariate analyses of factors influencing the occurrence of absences and the number of days of absence.

**Table 1 ijerph-19-12966-t001:** Study of compressible absences by professional and sociodemographic factors.

		Agents with ≥1 Absence/Year			*p*-Value between Groups	Effect Size between Groups with CI	Zero Inflated Negative Binomial (ZINB)	
Variables	n	n Agents		n Days of Absence			Incidence Risk Ratio (IRR)	
		(%)		Median (Range)			Count Model	Inflation Model
			Caregivers/non-caregivers					
Caregivers	4310	(39%)		13 (4 to 47)	**<0.001**	ref.	ref.	
Non-caregivers	12103	(44%)		16 (5 to 55)		**0.03 [0.01;0.04]**	1.19 ***	0.73 ***
			Professional category					
Medical	4861	(35%)		8 (2 to 24)	**<0.001**	ref.	ref.	
Paramedical	7640	(46%)		19 (6 to 64)		**0.27 [0.25;0.29]**	2.31 ***	0.64 ***
Administrative	2542	(37%)		12 (3 to 42)		**0.17 [0.15;0.19]**	1.96 ***	1.08 *
Technical	2713	(42%)		15 (5 to 53)		**0.28 [0.25;0.30]**	2.29 ***	0.82 ***
			Occupation					
Hospital practitioner	443	(31%)		8 (1 to 29)	**<0.001**	ref.	ref.	
Assistant nurse	2263	(50%)		20 (7 to 64)		**0.23 [0.18;0.27]**	1.89 ***	0.40 ***
Cleaners	1324	(54%)		20 (7 to 68)		**0.33 [0.28;0.37]**	2.36 ***	0.33 ***
Administrative officer	870	(46%)		13 (4 to 42)		**0.38 [0.33;0.43]**	2.44 ***	0.50 ***
			Professional status					
Tenured	8338	(27%)		19 (5 to 66)	**<0.001**	ref.	ref.	
Non-tenured	11739	(23%)		8 (3 to 24)		**0.32 [0.30;0.33]**	0.36 ***	1.56 ***
			Home-work distance					
<12 km	9040	(40%)		14 (4 to 45)	**<0.001**	ref.	ref.	
>12 km	6558	(44%)		17 (5 to 57)		**0.05 [0.04;0.06]**	1.04 *	0.76 ***
			Gender					
Male	4382	(36%)		11 (3 to 38)	**<0.001**	ref.	ref.	
Female	12031	(43%)		15 (5 to 52)		**0.09 [0.07;0.1]**	1.12 ***	0.66 ***
			Age					
<30 years old	8655	(40%)		10 (3 to 35)	**<0.001**	ref.	ref.	
30–40 years old	5506	(44%)		15 (5 to 48)		**0.15 [0.13;0.17]**	1.43 ***	1.01
40–50 year old	4705	(43%)		15 (5 to 48)		**0.21 [0.19;0.23]**	1.87 ***	1.00
>50 years old	4298	(44%)		18 (5 to 66)		**0.32 [0.30;0.34]**	2.60 ***	0.87 ***
			Marital status					
Single	8592	(24%)		10 (3 to 31)	**<0.001**	ref.	ref.	
Couple	8415	(26%)		17 (5 to 59)		**0.18 [0.17;0.19]**	1.58 ***	0.74 ***
Separated/widowed	1859	(29%)		20 (7 to 69)		**0.35 [0.33;0.37]**	2.03 ***	0.61 ***
			Parentality					
No children	9498	(39%)		12 (4 to 38)	**<0.001**	ref.	ref.	
≥1 children	8890	(44%)		16 (5 to 52)		**0.16 [0.15;0;17]**	1.48 ***	0.82 ***

* *p* value < 0.05, *** *p* value < 0.001.

**Table 2 ijerph-19-12966-t002:** Study of compressible absences by period and by professional and sociodemographic factors.

	Period 1—2007/2010				Period 2—2011/2014				Period 3—2015/2019		
	n	n(%) Agents with ≥1 Absence/Year	Median (Range)		n	n(%) Agents with ≥1 Absence/Year	Median (Range)		n	n(%) Agents with ≥1 Absence/Year	Median (Range)
Caregivers/non-caregivers											
Caregivers	6829	45	13 (4–43)		7051	41	18 (5–62)		8462	45	18 (5–61)
Non-caregivers	2808	46	8 (2–32)		2583	36	14 (4–49)		2655	36	19 (6–63)
Professional category											
Medical	1987	34	8 (4–17)		2000	29	10 (3–29)		2916	40	8 (1–25)
Paramedical	4842	47	14 (4–49)		5051	44	20 (6–68)		5546	46	21 (7–73)
Administrative	1484	44	6 (2–28)		1438	33	13 (4–46)		1504	34	18 (5–57)
Technical	1324	48	11 (3–38)		1145	40	15 (5–54)		1151	39	20 (6–75)
Occupation											
Hospital practitioner	247	33	7 (2–25)		270	23	13.5 (6–33)		318	34	6 (1–31)
Assistant nurse	1437	51	17 (5–55)		1488	49	22 (7–79)		1602	51	23 (8–79)
Cleaners	649	53	18 (7–57)		684	52	21 (7–80)		735	56	25 (8–88)
Administrative officer	499	53	7 (2–27)		461	41	14 (4–44)		420	44	21 (7–64)
Professional status											
Tenured	5719	29	13 (3–46)		6083	25	20 (6–71)		6433	26	25 (8–81)
Non-tenured	3918	23	8 (3–22)		3551	20	9 (3–29)		4684	25	7 (2–22)
Home-work distance											
<12 km	4368	46	11 (3–39)		5087	37	15 (5–55)		6361	40	16 (5–55)
>12 km	3605	48	12 (3–46)		4139	42	18 (5–61)		4643	45	21 (6–69)
Gender											
Male	2414	42	9 (2–31)		2324	34	13 (4–47)		2878	36	14 (4–48)
Female	7223	46	12 (4–42)		7310	41	18 (5–61)		8239	44	19 (6–64)
Age											
<30 years old	3566	39	10 (4–31)		3520	38	11 (3–43)		4673	42	10 (3–33)
30–40 years old	2133	46	12 (3–41)		2158	41	16 (5–53)		2372	44	18 (5–58)
40–50 year old	2121	47	11 (3–39)		2017	39	17 (6–56)		2046	42	21 (7–69)
>50 years old	1817	50	13 (3–50)		1939	41	22 (7–96)		2026	42	28 (9–94)
Marital status											
Single	3735	25	9 (3–25)		3676	21	10.5 (3–33)		4933	26	11 (3–33)
Couple	5018	28	12 (3–43)		5054	25	19 (6–65)		5298	26	21 (6–70)
Separated/widowed	855	31	14 (5–50)		859	28	21 (7–73)		812	29	25 (9–84)
Parentality											
No children	4237	27	10 (4–34)		4222	24	14 (4–48)		5452	27	12 (3–39)
≥1 children	5400	29	12 (3–42)		5412	25	18 (5–62)		5665	26	22 (7–71)

## Data Availability

Data cannot be shared publicly due to confidentiality. Data are available from the Institutional Data Access/Ethics Committee of the Clermont-Ferrand University Hospital (contact via the Human Resources Department) for researchers who meet the criteria for access to confidential data.

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
