# Peer review of "Statistical Analysis of Absenteeism in a University Hospital Center between 2007 and 2019"

_ijerph, 2022, doi:10.3390/ijerph191912966_

Round 1
Reviewer 1 Report (Previous Reviewer 1)
The manuscript has been improved includiong the limitation section.
Reviewer 2 Report (Previous Reviewer 2)
The authors addressed all my suggestions. Some lines are displayed in red (eg. 122-124). You should remove the red color and let the manuscript in black only.
Reviewer 3 Report (Previous Reviewer 3)
I thank the authors for clarifying all the issues in the previous version of the manuscript.
This manuscript is a resubmission of an earlier submission. The following is a list of the peer review reports and author responses from that submission.
Round 1
Reviewer 1 Report
This study addresses a topic of relevant importance such as absenteeism in the workplace which also represents an impact on public health. however, there are numerous critical issues. 1. The study seems to be anachronistic if we consider how today the advent of the pandemic has had an impact on the world of work, especially in the health sector with an increasing number of studies about absenteeism due to COVID-19 (e.g. Edge R, van der Plaat DA, Parsons V, et al. Changing patterns of sickness absence among healthcare workers in England during the COVID-19 pandemic. J Public Health 2021. doi:10.1093/pubmed/fdab341; Ahmad Faramarzi, Javad JavanNoughabi, Seyed Saeed Tabatabaee, Ali Asghar Najafpoor & Aziz Rezapour. The lost productivity cost of absenteeism due to COVID-19 in health care workers in Iran: a case study in the hospitals of Mashhad University of Medical Sciences. BMC Health Services Research volume 21, Article number: 1169 (2021); Basem Gohar. Sickness absence in healthcare workers during the COVID-19 pandemic. Occupational Medicine, Volume 70, Issue 5, July 2020, Pages 338–342). COVID-19 changed the meaning of absenteeism in companies, the causes of absenteeism, and the way absences were managed during this crisis. Teleworking or virtual offices contributed to this fact, making absenteeism harder to record. This had an important impact on businesses, whether they had used teleworking in the past or not (Organisation for Economic Co-operation and Development (OECD) (2020). Supporting people and companies to deal with the COVID-19 virus. OECD Policy Responses to Coronavirus. Retrieved from http://www.oecd.org/coronavirus/policyresponses/supportingpeople-and-companies-to-deal-with-the-covid-19-virusoptions-for-an-immediate-employmentand-social-policy-response-d33dffe6/) 2. To estimate the evolution of compressible absenteeism in a hospital center, the study takes into account only the professional and sociodemographic factors that influence absenteeism, but workplace absenteeism is a complex issue that can be caused by other factors which include job satisfaction, organizational environment, job demand, relazioni al lavoro, supporto, Support from colleagues and support from supervisors Health at work, workload, psychosocial risk, etc 3. Without considering all the factors wich can cause abseenteism at work is not possible to propose adapted and personalized preventive measures.
Reviewer 2 Report
Lines 64 to 66 read: "Also, publications dealing specifically with absenteeism in health care institutions are generally limited to a few years of study only (10)." some more details should be presented regarding the existing literature on absenteeism in health care institutions. This way, the introduction would be more comprehensive.
Conclusions should reiterate the value added of the study.
Reviewer 3 Report
Review:
ijerph-1848178
Title:
Statistical analysis of absenteeism in a University Hospital
Center between 2007 and 2019
Authors:
Charlène Millot, Bruno Pereira, Sophie Miallaret, Maëlys Clinchamps, Luc Vialatte, Arnaud Guillin,
Yan Bailly, Ukadike Chris Ugbolue, Valentin Navel, Julien Steven Baker, Jean-Baptiste Bouillon-Minois,
Frédéric Dutheil.
This paper deals with important topics that may have relevant implications in hospital management and to implement preventive measures. I congratulate the authors for facing the endeavor. A total of 16,413 employees were included in this study over 12 years: a very good starting point. Nevertheless the analysis omitted to consider key features in these data that make the conclusions on p-values almost a matter of faith.
The authors are free to limit their analyses to the discussion of descriptive statistics, which might lead anyway to very important conclusions and information, otherwise inference about the reference population of workers needs improvements.
I regret to inform you that in my opinion this paper can't be accepted in its current form. Below I am going to list major concerns about the statistical analysis with the aim of suggesting some possible lines of work during the revision.
(1) Autocorrelation.
In Section 2.1 of the manuscript it has been stated that: "In agreement with the CNIL, a unique 6-digit identifier was assigned to each university hospital center employee to ensure the anonymity of data." This means that, despite anonymity, the 12 consecutive years are available for the analysis with information at elementary level on each worker, i.e. without any aggregation. Nevertheless, the autocorrelation of random variables pertaining to the same statistical unit (worker) along time has been neglected.
Textbooks delaing with the analysis of panel data include "Principles of Econometrics", by R. Carter Hill, William E. Griffiths and Guay C. Lim, chapter.
In other words, a worker with high rate of absence in the first 3 years could be expected to have high rates in the next 5-9 years? A worker with low rate of absence in the first 3 years could be expected to have low rates in the next 5-9 years? I believe both answers are Yes. Did you check this?
(2) Normality.
Page 2 Rows 106-107. It seems that the authors performed a statistical test on the marginal distribution of the response variable and then declared that it does not follow the normal distribution. Well, even in multiple regression with Normal residuals the marginal distribution is typically not normal: residuals have to be normal, not the marginal distribution of the outcome Y. Please clarify this point.
(3) p-values - applicative importance
Most of the conclusions based on models rests on p-values. Given the huge size of this sample (more than 1M observations) it is expected that the null hypothesis may be OFTEN rejected, but it is not clear if statistical significance matches the significance in the considered domain of application. Please comment more on this point.
(4) p-values - several tests
What about the familywise error rate for tens statisical tests that were performed? Was any protection considered?
(5) p-values - inappropriate calculations
Authors honestly recognized that even the ZINB distribution is not suited for these analyses, but it then follows that those pvalues are not correct, and the conclusions based on them should be considered not trustable.
(6) Median instead of the mean.
The authors considered the median as summary because of their claim that normality was not appropriate. The motivation is weak because of points (1) and (2) above, and because Bootstrap resampling techniques make possible to use the mean anyway, by producing an approximation of the sampling distribution without assuming normality. Well, Boostrap could be also applied after choosing the median instead of the mean as sample statistic, see for example
Divison and Hinkley, 2013, Bootstrap Methods and their Application, Cambridge.
(7) Limitations and extensions
I congratulate with the authors for clearly declaring that the ZINB family has not been fully satisfactory. Honesty deserves attention. Nevertheless it is not clear what did prevent the use of mixture models to extend the ZINB, maybe just with a second component addressing the deficiency in the right tail of the distribution.
Please see McLachlan, Peel, 2000, Finite Mixture Models, Wiley.
But once more, Boostrap methods could work well even in this case.
Last but not least, a Bayesian mixture model could also be considered,
chapter 22